# The Vital Role of La_2_O_3_ on the La_2_O_3_-CaO-B_2_O_3_-SiO_2_ Glass System for Shielding Some Common Gamma Ray Radioactive Sources

**DOI:** 10.3390/ma14174776

**Published:** 2021-08-24

**Authors:** Hanan Al-Ghamdi, Mengge Dong, M. I. Sayyed, Chao Wang, Aljawhara H. Almuqrin, Fahad I. Almasoud

**Affiliations:** 1Department of Physics, College of Science, Princess Nourah Bint Abdulrahman University, Riyadh 11671, Saudi Arabia; hmalghmdi@pnu.edu.sa (H.A.-G.); ahalmoqren@pnu.edu.sa (A.H.A.); 2Department of Resources and Environment, School of Metallurgy, Northeastern University, Shenyang 110819, China; mg_dong@163.com; 3Department of Nuclear Medicine Research, Institute for Research and Medical Consultations (IRMC), Imam Abdulrahman Bin Faisal University (IAU), P.O. Box 1982, Dammam 31441, Saudi Arabia; 4Department of Mechanical Engineering, The University of Texas at Dallas, Richardson, TX 75080, USA; Wang.Chao@UTDallas.edu; 5Nuclear Science Research Institute (NSRI), King Abdulaziz City for Science and Technology (KACST), Riyadh 11442, Saudi Arabia; fmasaud@kacst.edu.sa; 6Department of Soil Sciences, College of Food and Agricultural Sciences, King Saud University, Riyadh 12372, Saudi Arabia

**Keywords:** Phy-X software, radiation attenuation, linear attenuation coefficient, La_2_O_3_

## Abstract

The role La_2_O_3_ on the radiation shielding properties of La_2_O_3_-CaO-B_2_O_3_-SiO_2_ glass systems was investigated. The energies were selected between 0.284 and 1.275 MeV and Phy-X software was used for the calculations. BLa10 glass had the least linear attenuation coefficient (*LAC*) at all the tested energies, while BLa30 had the greatest, which indicated that increasing the content of La_2_O_3_ in the BLa-X glasses enhances the shielding performance of these glasses. The mass attenuation coefficient (*MAC*) of BLa15 decreases from 0.150 cm^2^/g to 0.054 cm^2^/g at energies of 0.284 MeV and 1.275 MeV, respectively, while the *MAC* of BLa25 decreases from 0.164 cm^2^/g to 0.053 cm^2^/g for the same energies, respectively. At all energies, the effective atomic number (*Z_eff_*) values follow the trend BLa10 < BLa15 < BLa20 < BLa25 < BLa30. The half value thickness (*HVL*) of the BLa-X glass shields were also investigated. The minimum *HVL* values are found at 0.284 MeV. The *HVL* results demonstrated that BLa30 is the most space-efficient shield. The tenth value layer (TVL) results demonstrated that the glasses are more effective attenuators at lower energies, while decreasing in ability at greater energies. These mean free path results proved that increasing the density of the glasses, by increasing the amount of La_2_O_3_ content, lowers *MFP*, and increases attenuation, which means that BLa30, the glass with the greatest density, absorbs the most amount of radiation.

## 1. Introduction

Radiation protective materials (sometime called shielding materials) are used widely in diagnostic radiology, nuclear facilities and hospitals to decrease the exposure of the ionizing radiations to medical physicist, patients and workers on the nuclear facilities. The main goal of the radiation shielding is to protect the persons who are using the X-ray equipment from the harmful effects of the photons that are emitted from these equipment. It is worth determining the type, thickness and composition of the shielding materials that can be used practically in real applications [1,2]. Glasses are utilized in the windows of the X-ray rooms and other constructions due to the amazing features of the glasses, such as the transparency of the light, low fabrication cost, the simplicity of the preparation method, and most importantly, the glasses show better radiation protection characteristics in comparison with other materials such as polymers, ceramics and concretes [3,4,5,6]. Many recent research has shown that the radiation shielding attitudes for glassy medium can be enhanced by changing the chemical composition and adding a certain amount of heavy metal oxides (HMO) and this is a very easy task during the preparation of the glasses [7,8,9]. Moreover, the recent works proved that the glasses with HMO are preferable in practical applications than using thick concrete for many reasons such as the concrete is an opaque and this limited the utilizations of the concrete in some applications [10,11]. Among several types of glasses, B_2_O_3_-based glasses have a wide range of commercial applications thanks to the good physical features like low melting temperature, high transparency and high thermal stability. Moreover, B_2_O_3_ glasses can be prepared easily with HMO and these HMO are mainly added to the borate glasses for increasing the density of the prepared glass system and thus to enhance the radiation shielding attitudes [12,13,14]. In the lanthanide series, La (Z = 57) is the first element in this series. Lanthanum oxide is an inorganic compound which has a very high dielectric constant and was utilized in fabricating certain types of optical glasses like telescope lenses and cameras. Moreover, it is used in some ferroelectric materials. The beneficial properties of La are of a high atomic number and density (the density of La_2_O_3_ is around 6.5 g/cm^3^). In addition to this, La_2_O_3_ glasses are colorless, which allows these glasses to be used in typical applications as radiation shielding windows or eyeglasses [15]. For these reason, we aim in this work to examine the influence of the addition of La_2_O_3_ in borosilicate glasses on radiation shielding properties.

## 2. Materials and Methods

The glass systems selected in this investigation have the composition of La_2_O_3_-CaO-B_2_O_3_-SiO_2_ and were fabricated by Kaewjaeng et al. [16] Five glass samples were labelled according to the content of La_2_O_3_ as follows:

BLa10: 10 La_2_O_3_-10 CaO-70 B_2_O_3_-10 SiO_2_, ρ = 3.01 g/cm^3^

BLa15: 15 La_2_O_3_-10 CaO-65 B_2_O_3_-10 SiO_2_, ρ = 3.16 g/cm^3^

BLa20: 20 La_2_O_3_-10 CaO-60B_2_O_3_-10 SiO_2_, ρ = 3.40 g/cm^3^

BLa25: 25 La_2_O_3_-10 CaO-55B_2_O_3_-10 SiO_2_, ρ = 3.60 g/cm^3^

BLa30: 30 La_2_O_3_-10 CaO-50B_2_O_3_-10 SiO_2_, ρ = 3.76 g/cm^3^

These compositions are in mol%. As we can see, the density of the BLaX glasses increases from 3.01 g/cm^3^ to 3.76 g/cm^3^ due to the increase in the La_2_O_3_ from 10 to 30 mol%. This is expected increment in the density, since the low molecular weight compound (i.e., B_2_O_3_) is replaced by the heavier one (i.e., La_2_O_3_). Therefore, the glass becomes denser with the addition of La_2_O_3_. It is worth mentioning that the density of the glass plays a very vital role in the shielding ability for these BLaX glasses. In the current investigation, we will examine the photons attenuation factors of the selected BLa5-BLa25 glasses by Phy-X software [17]. This is a very useful and quick software that can evaluate several radiation shielding factors for any compounds, mixtures, glasses, etc., at different energy values. The calculations of the attenuation factors using the software can be summarized in 3 steps as follows: (a) definition of the materials (by adding the composition and the density); (b) selection the energy; and (c) chosen the factors to be evaluated. It is important to mention that in the first step, the use has two choices for the definition of the materials, namely either in mol% or in w.t %. Moreover, the density must be defined in the unit of g/cm^3^. As an example for the first step, the BLa10 glass is defined in the main screen interface of this program (available on https://phy-x.net/PSD by: 10 La_2_O_3_ + 10 CaO + 70 B_2_O_3_ + 10 SiO_2_. In the current work, we chose 13 energies (ranging from 0.122 to 1.458 MeV).

We determined the basic radiation shielding parameters for the La_2_O_3_-CaO-B_2_O_3_-SiO_2_ glasses. One of these parameters is the mass attenuation coefficient (*MAC*) [18,19,20]. For a certain medium, the *MAC* simply describes how easily it can be penetrated by a beam of photons. For a medium composed of different elements (such as our samples in this work), the mixture rule is useful in determining the *MAC* and can be written as:(1)$$MAC=\underset{i}{\sum }w_{i}{\left(MAC\right)}_{i}$$
where the symbol w denotes the weight fraction, and the (*MAC*)_i_ is the mass attenuation coefficient of the *i*th constituent element.

If we take the density of the medium into the account, then we need to determine the linear attenuation coefficient (*LAC*), which is given by:*LAC* = *MAC* × density(2)

The last parameter is helpful in evaluating the half value layer (*HVL*) and the mean free path (*MFP*). The *HVL* gives information about the thickness of the medium which can attenuate half of the incoming photons, while, the *MFP* gives information about the average distance traveled by the photon between two collisions. Both parameters are related to the *LAC* by Equations (3) and (4) [21]:(3)$$HVL=\frac{0.693}{LAC}$$
(4)$$MFP=\frac{1}{LAC}$$

The effective atomic number (*Z_eff_*) can be calculated using the following formula [22]:(5)$$Z_{eff}=\frac{\sum_{i}f_{i}A_{i}{\left(MAC\right)}_{i}}{\sum_{j}f_{j}{\frac{A_{j}}{Z}}_{i}{\left(MAC\right)}_{j}}$$
where *f_i_* is the fraction by mole of the each constituent element and *A_i_* is the atomic weight.

## 3. Results

The *LAC* of the BLa-X glasses was determined and plotted at selected energies in Figure 1. The *LAC* of a sample describes the attenuation ability of the material, with a greater value meaning more attenuation. The energies in this investigation were selected as: 0.284, 0.3645, 0.511, 0.637, 0.6617, 0.723 and 1.275 MeV. Two general trends can be observed in the figure. First, the *LAC* values for all five BLa-X glasses decrease with increasing energy [18]. This decrease in value is due to the increased penetration power of higher energy photons, reducing the attenuation of the samples, and decreasing *LAC*. The second trend of the figure can be seen by analyzing the values at a single energy. As can be observed, BLa10 has the least *LAC* at all the tested energies, while BLa30 has the greatest. This indicates that the shielding ability for the BLa-X glasses is enhanced when the La_2_O_3_ changes from 10 to 30 mol%. This is because the LA has much higher atomic number than that of B. Therefore, increasing the amount of La_2_O_3_ at the expense of B_2_O_3_ causes an enhancement in the density value, and this explained the increase in the *LAC* with more La_2_O_3_ added to the glasses. This observation can be confirmed by looking at the positive correlation between La_2_O_3_ mol% and the *LAC* of the glasses (see Figure 2).

Figure 3 shows the *MAC* of the BLa-X glasses against energy. Like *LAC*, a greater *MAC* represents a more radiation shielding efficient shield [19]. Unlike *LAC*, however, *MAC* factors out density to compare the glasses. The same energies were chosen as the previous figures. Even though the two parameters are alike, Figure 3 demonstrates an interesting trend between *MAC* and energy. *MAC* also decreases with increasing energy for the same reason as previously stated. The *MAC* of BLa15 decreases from 0.150 to 0.054 cm^2^/g at energies of 0.284 and 1.275 MeV, respectively, while the *MAC* of BLa25 decreases from 0.164 to 0.053 cm^2^/g for the previously mentioned energies, respectively. Clearly, at higher energies, the difference between the *MAC* is almost negligible, and this is explained according to the Compton scattering as stated in [23]. At 1.275 MeV, BLa10 has an *MAC* equal to 0.054 cm^2^/g, while BLa30 has a *MAC* equal to 0.053 cm^2^/g. These results contrast with the trend observed in Figure 1, where BLa10 had the least *MAC* at all energies. The trend in this figure demonstrates that at low energies, BLa30 remains the most desirable shield, but at greater energies, there is almost no difference between the attenuation abilities of the glasses, with BLa10 having a slightly greater advantage due to the slightly higher *MAC* of BLa10 than the other glasses. The reason for the very slight difference in the *MAC* for the selected glasses between 0.637 and 1.275 MeV is related directly to the Compton scattering, as it is known that this process has a small dependence on the atomic number of the medium.

The *Z_eff_* of the BLa-X glasses are plotted in Figure 4. The *Z_eff_* values follow the trend BLa10 < BLa15 < BLa20 < BLa25 < BLa30. In this figure, BLa30 clearly has the greater *Z_eff_* at all energies. This trend is due to BLa30 having the greatest La_2_O_3_ content, which has a greater atomic number than B_2_O_3_, which it is being substituted by. This conclusion also explains why BLa10 has the least *Z_eff_* at all energies. Nevertheless, all the *Z_eff_* values decrease with the increasing energy. At 0.284 and 0.365 MeV, the *Z_eff_* is sharply decreased. This sharp trend is thanks to the dominance of the photoelectric effect at these two energies. As energy further increases, the Compton scattering becomes the most important process. At this medium energy range, the *Z_eff_* can be observed to only slightly decrease. Despite this decrease, BLa30 maintains the highest *Z_eff_* at all chosen energies.

The *HVL* of the BLa-X glass shields were investigated at selected energies and in Figure 5. One must remember that a lower *HVL* signifies a more space-efficient shield. The minimum *HVL* values are found at 0.284 MeV, which is as expected. Figure 5 demonstrated that BLa30 has the least *HVL* and BLa10 has the greatest, which means that BLa30 is the most space-efficient shield. As energy increases, the *HVL*s of the glasses increase as well. The *HVL* of BLa10 increases from 1.646 (for E = 0.284 MeV) to 4.268 cm (for E = 1.275 MeV), but it is increased for BLa30 from 1.083 to 3.499 cm for the same respective energies. Since energy and *HVL* are directly correlated, the maximum *HVL* can be found at the highest selected energy, 1.275 MeV. This positive trend occurs since, as the photon energy increases, the dimension of the medium required to shield 50% of the radiation increases, increasing the *HVL*. Even with the increase in values, however, BLa30 has the least *HVL*.

Mean free path represents the distance between subsequent collisions between the incoming photons and the atoms within the sample. A lower *MFP* signifies that the distance between collisions is short, increasing the number of total collisions, and increasing the amount of attenuation. Therefore, a smaller *MFP* is more desirable. Figure 6 graphs the *MFP* of the samples against the density of the five glasses at selected energies. At any single energy, an inverse correlation can be observed between density and *MFP*. This decrease in value demonstrates that increasing the density of the glasses, by increasing the amount of La_2_O_3_ content, lowers *MFP*, and increases attenuation, which means that BLa30, the glass with the greatest density, absorbs the most amount of radiation. This conclusion can also be drawn by analyzing any of the other densities. By the same logic, the glass with the least density—BLa10—has the greatest *MFP* at all energies. The *MFP* values can also be seen to increase with the increasing energy. An especially large change in values occurs between 0.72 MeV and 1.28 MeV, which shows how the glasses decrease in efficiency as the photon energy increases. This is because the penetrating ability of the high energy photons is very high and thus we needed a relatively thick glass to attenuate these high energy photons. With this in mind, to maximize attenuation, the BLa-X glass with the greatest density should be used at low energies.

To put the shielding ability of the BLa-X glasses into perspective, the *HVL* of the tested samples were compared against three commercially available glasses, RS-360, RS-520, and RS-253-G18, in Figure 7. It should first be stated that the *HVL* of all the glasses increases with increasing energy, even the commercially available glasses. The same trend can be observed at all three energies, following the order of RS-253-G18 > BLa10 > BLa15 > BLa20 > BLa25 > BLa20 > RS-360 > RS-520. The high *MFP* for the RS-253-G18 is due to the low density of this glass compared to our selected glasses, where the density of RS-253-G18 is 2.53 g/cm^3^, while the density of our glasses varied between 3.01 and 3.76 g/cm^3^. This figure shows that all the BLa-X glasses are more effective than RS-253-G18, but they are all outperformed by RS-360 and RS-520 at low, medium, and high energies. It can be observed, however, that at 1.275 MeV BLa30 has comparable attenuation to RS-360, although RS-520 has a much smaller *MFP* than all the other tested samples.

## 4. Conclusions

The linear attenuation coefficient (*LAC*) of the BLa-X glasses was reported to be between 0.284 and 1.275 MeV. The *LAC* values for all five BLa-X glasses decrease with the increasing energy, indicating the reduction in the attenuation of the samples. At 0.284 MeV, the *LAC* values varied between 0.421 cm^−1^ for BLa10 and 0.640 cm^−1^ for BLa30. Whereas, at 1.275 MeV, the *LAC* values varied between 0.162 and 0.198 cm^−1^ for BLa10 and BLa30, respectively. BLa10 has the least *LAC* as well as *MAC* at all the tested energies, while BLa30 has the greatest. At 0.511 MeV, the *Z_eff_* values varied between 10.48 and 16.43. The *MAC* of BLa15 decreases from 0.150 cm^2^/g to 0.054 cm^2^/g at energies of 0.284 MeV and 1.275 MeV, respectively, while the *MAC* of BLa25 decreases from 0.164 cm^2^/g to 0.053 cm^2^/g. The minimum *HVL* values are found at 0.284 MeV, and they are equal to 1.646, 1.460, 1.288, 1.168, and 1.083 for BLa10, BLa15, BLa20, BLa25, and BLa30, respectively. BLa30 has the least TVL and *HVL*, while BLa10 has the greatest, which emphasized the positive correlation between La_2_O_3_ content and shielding ability, as well as the conclusion that BLa30 has the most potential for radiation shielding applications. Moreover, an inverse correlation was observed between density and *MFP* and from the *MFP* results, we found that BLa30, the glass with the greatest density, absorbs the most amount of radiation. All the BLa-X glasses are more effective than RS-253-G18, but they are all outperformed by RS-360 and RS-520 at low, medium, and high energies. Finally, at 1.275 MeV, BLa30 has comparable attenuation to RS-360.

## Figures and Tables

**Figure 1 materials-14-04776-f001:**
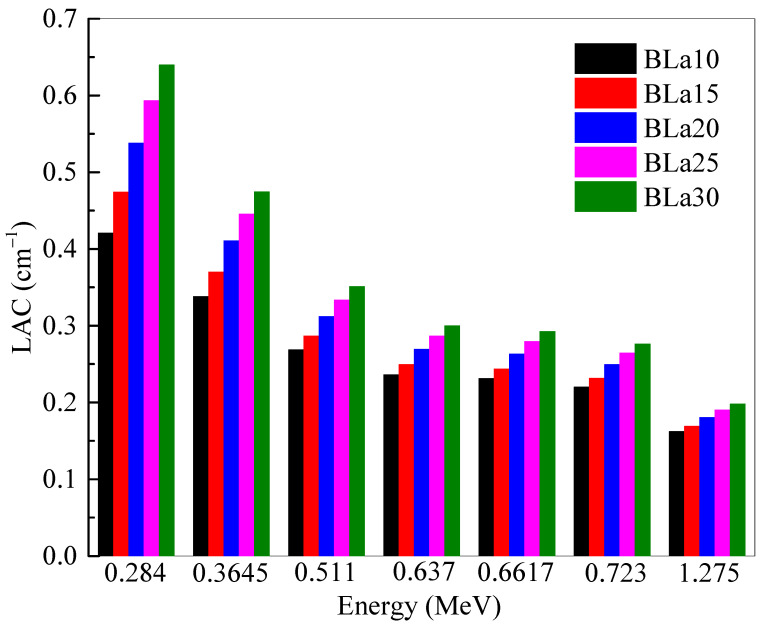
Linear attenuation coefficient of the shielding samples at 0.284, 0.3645, 0.511, 0.637, 0.6617, 0.723 and 1.275 MeV.

**Figure 2 materials-14-04776-f002:**
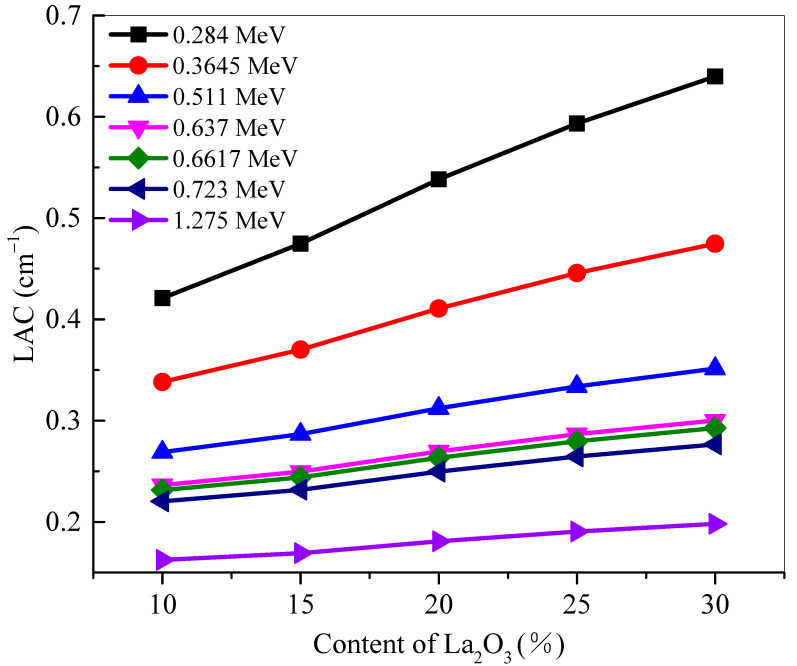
Effect of the content of La_2_O_3_ on the shielding performance for samples at 0.284, 0.3645, 0.511, 0.637, 0.6617, 0.723 and 1.275 MeV.

**Figure 3 materials-14-04776-f003:**
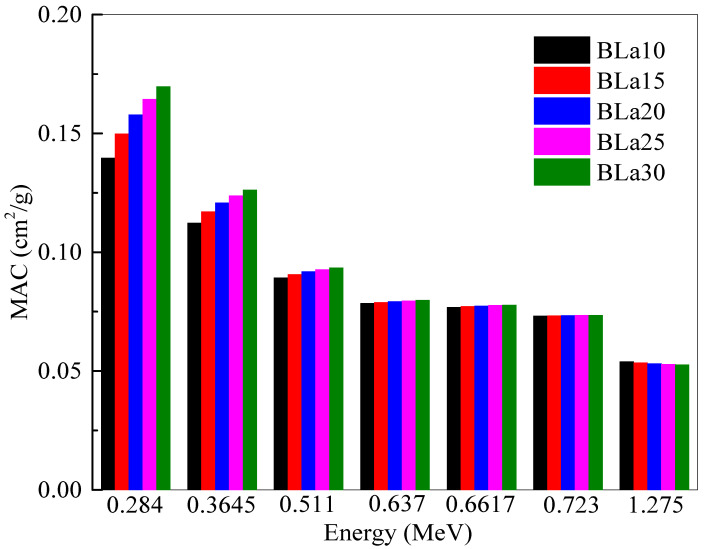
Mass attenuation coefficient (*MAC*) of the chosen samples at 0.284, 0.3645, 0.511, 0.637, 0.6617, 0.723 and 1.275 MeV.

**Figure 4 materials-14-04776-f004:**
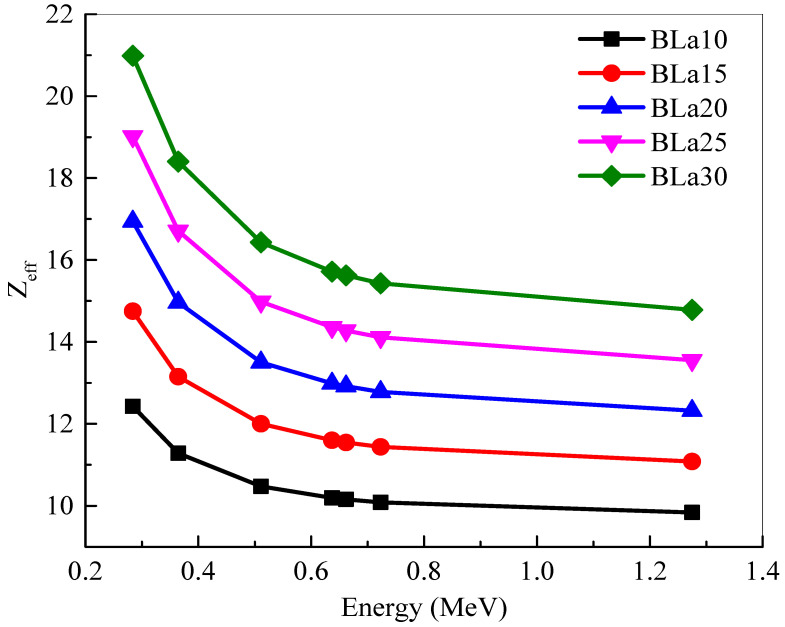
Effective atomic number (*Z_eff_*) of the shielding samples at 0.284, 0.3645, 0.511, 0.637, 0.6617, 0.723 and 1.275 MeV.

**Figure 5 materials-14-04776-f005:**
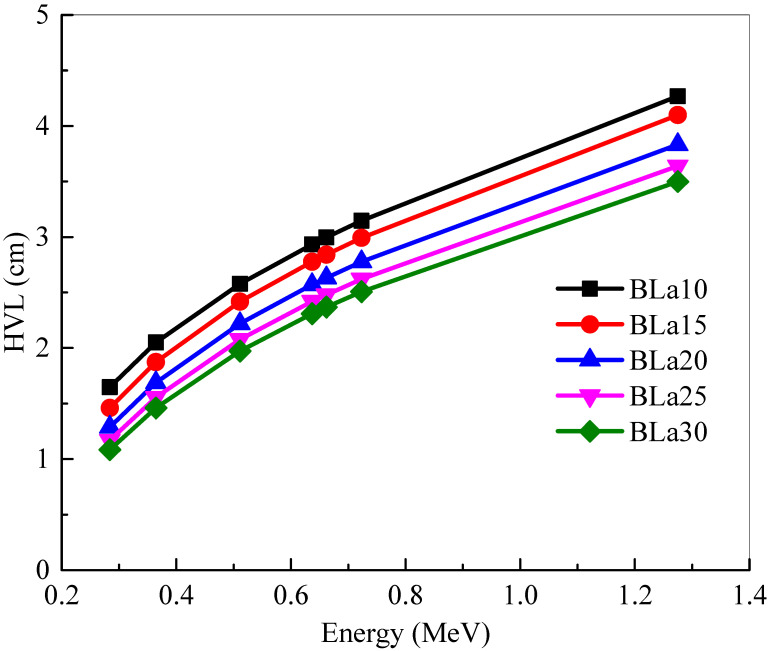
Half value thickness (*HVL*) of the shielding samples at 0.284, 0.3645, 0.511, 0.637, 0.6617, 0.723 and 1.275 MeV.

**Figure 6 materials-14-04776-f006:**
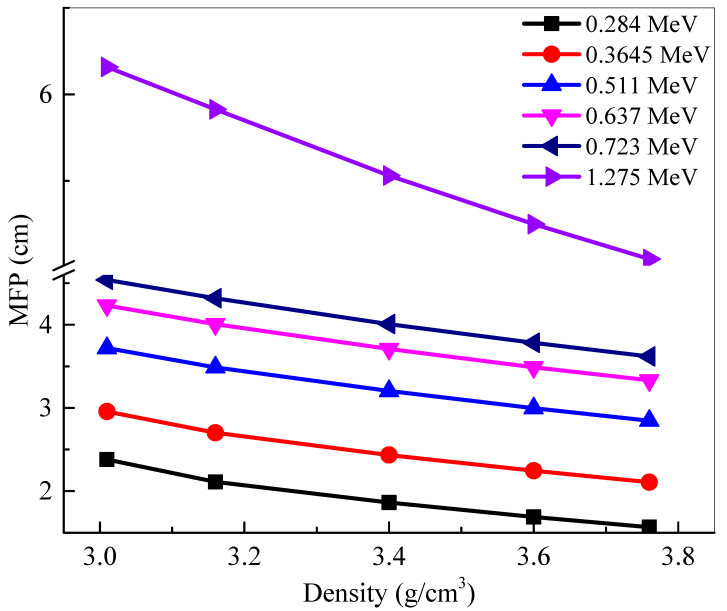
The mean free path (*MFP*) of the shielding samples vs. the density.

**Figure 7 materials-14-04776-f007:**
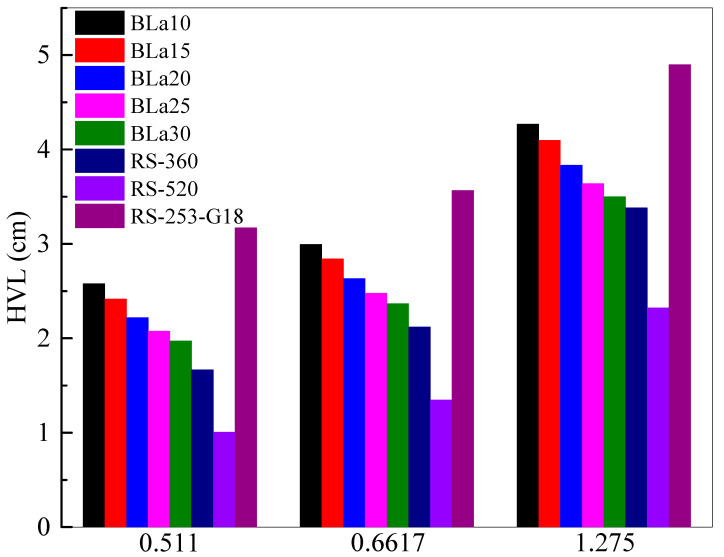
Comparison with the commercial shielding glasses.

## Data Availability

The data presented in this study are available on request from the corresponding author.

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
