# Peer review of "The Vital Role of La2O3 on the La2O3-CaO-B2O3-SiO2 Glass System for Shielding Some Common Gamma Ray Radioactive Sources"

_materials, 2021, doi:10.3390/ma14174776_

Round 1
Reviewer 1 Report
This manuscript describes an experimental study, aided by theoretical information, of the absorption properties of various glasses pertaining to high energy photons. I have previously
reviewed a similar study by some of the same authors for other types of glasses. That manuscipt contains information necessary for understanding how the present results were obtained,
e.g. how the LAC and MAC values were obtained and how XCOM software were used in the analysis. No reference to that manuscript is made; therefore essential information is
not available to the reader. Assuming this will be corrected, the present information is, in general, well presented but somewhat "padded" with respect to the text, e.g., numerical
values are provided but could easily be extracted from the figures. Also, essentially the same information is provided over and over, just in slightly different ways, e.g., Figs. 1, 5,6, and 8 are
all provide basically the same information. Also, two types of formats are used, e.g., histogram format and standard line+symbol format. In some cases, e.g., Fig 1, the histogram format
works well with what the authors want to emphasize, but in others, e.g., Fig 6, I don't understand why the Fig 5 format isn't used.
My overall opinion is that this is a well performed and analyzed study that is worthy of publication (hopefully it isn't followed by similar studies which could have been combined
into a single manuscript). But, prior to publication, the items mentioned above and below should be considered in order to emphasize the main points in a more efficient manner.
abstract, line 29: The sentence beginning with BLa30 is simply a repeat of information in the previous sentence and can be deleted.
section 2, materials and methods: Please provide brief explanations, e.g., a couple sentences in each case plus a single equation if necessary, of how the LAC, MAC
were determined.
page 5, Zeff discussion: It is stated that Zeff is the average atomic number of a particular sample. If so, I do not understand how this value has an energy dependence as seen in
Fig. 4.
Figures 5-8 and associated discussion: The essential information of these figures is that 1) the radiation penetration increases with energy and 2) decreases for higher Z glasses.
This has already been shown in figures 1-4. Thus figures 5-8 are simply showing the same thing in different ways. However, some of this information is useful, but could be presented
in a much more compact form, e.g. eliminate most of the text, in particular any refererence to particular numberical values, and provide just a brief statement of the various properties
being shown. Second, show figure 6 using the same style as in figure 5. Third, is figure 8 necessary since figure 1 already has shown the LAC? Four: Summarize the findings at the
end of this section or in the conclusion.
Figure 9: Very nice. This is probably the message that is most important to the reader.
Author Response
Comments and Suggestions for Authors
This manuscript describes an experimental study, aided by theoretical information, of the absorption properties of various glasses pertaining to high energy photons. I have previously
reviewed a similar study by some of the same authors for other types of glasses. That manuscipt contains information necessary for understanding how the present results were obtained,
e.g. how the LAC and MAC values were obtained and how XCOM software were used in the analysis. No reference to that manuscript is made; therefore essential information is
not available to the reader. Assuming this will be corrected, the present information is, in general, well presented but somewhat "padded" with respect to the text, e.g., numerical
values are provided but could easily be extracted from the figures. Also, essentially the same information is provided over and over, just in slightly different ways, e.g., Figs. 1, 5,6, and 8 are
all provide basically the same information. Also, two types of formats are used, e.g., histogram format and standard line+symbol format. In some cases, e.g., Fig 1, the histogram format
works well with what the authors want to emphasize, but in others, e.g., Fig 6, I don't understand why the Fig 5 format isn't used.
My overall opinion is that this is a well performed and analyzed study that is worthy of publication (hopefully it isn't followed by similar studies which could have been combined
into a single manuscript). But, prior to publication, the items mentioned above and below should be considered in order to emphasize the main points in a more efficient manner.
Reply: the authors would like to thank you very much for these constructive comments which improve our paper, we read these comments carefully and tried to follow your comments and we revised our paper accordingly.
Regarding to your comment that this is similar study to some of the same authors for other types of glasses: We would like to explain that this is a different paper and it is a continuing effort of researchers like us to find suitable cheaper glass compositions with possible high density for gamma-ray shielding. The glasses can be manufactured with numerous combinations from former to base structure as well as additive types. We would like to mention that our papers and ongoing projects are NOT arbitrary chemical composition changes. For example, there are thousands of composites and alloy materials exist. Each one has a different structure. Investigating their radiation attenuation competencies does not mean that researchers perform the same study. Of course, our methodology is the same since the terms of shielding parameters are the same. We believe as a researcher you know this. Which parameters else can be applied to study radiation shielding? We should follow the proven and confirmed shielding parameters which widely reported by other researchers.
For your other valuable comments, we revised the paper according to these comments as you will see in the next points.
abstract, line 29: The sentence beginning with BLa30 is simply a repeat of information in the previous sentence and can be deleted.
Reply: Thank you very much for your comment, we agree with you and deleted this sentence
section 2, materials and methods: Please provide brief explanations, e.g., a couple sentences in each case plus a single equation if necessary, of how the LAC, MAC
were determined.
Reply: Thank you very much for your comment, we agree with you and extended the materials and methods, and added a brief explanations about each parameter and provided the equations used in the calculation of the MAC, LAC, HVL, MFP, TVL and Zeff.
page 5, Zeff discussion: It is stated that Zeff is the average atomic number of a particular sample. If so, I do not understand how this value has an energy dependence as seen in
Fig. 4.
Reply: Thank you for the comment, we deleted this sentence since we feel that it is not accurate, while the Zeff has an energy dependence and you can refer for the following papers for more details about this point:
-Berna Oto, Nergiz Yıldız, Fatma Akdemir, Esra Kavaz, Investigation of gamma radiation shielding properties of various ores, Progress in Nuclear Energy 85 (2015) 391-403
Vishwanath P. Singh, N.M. Badiger, J. Kaewkhao, Radiation shielding competence of silicate and borate heavy metal oxide glasses: Comparative study, Journal of Non-Crystalline Solids 404 (2014) 167–173
- Singh, H. Singh, V. Sharma, R. Nathuram, A. Khanna, R. Kumar, S.S. Bhatti, H. Sahota, Nucl. Inst. Methods Phys. Res. B 194 (2002) 1–6.
S Yasmin, ZS Rozaila, MU Khandaker, BS Barua, FUZ Chowdhury, MA Rashid, DA Bradley, The radiation shielding offered by the commercial glass installed in Bangladeshi dwellings, Radiation Effects and Defects in Solids 173 (7-8), (2018) 657-672
- I. Sayyed, Faras Q. Mohammed, K. A. Mahmoud, Eloic Lacomme, Kawa M. Kaky, Mayeen Uddin Khandaker, Mohammad Rashed Iqbal Faruque.2020, Evaluation of Radiation Shielding Features of Co and Ni-Based Superalloys Using MCNP-5 Code: Potential Use in Nuclear Safety, Appl. Sci. 2020, 10, 7680; doi:10.3390/app10217680
Figures 5-8 and associated discussion: The essential information of these figures is that 1) the radiation penetration increases with energy and 2) decreases for higher Z glasses.
This has already been shown in figures 1-4. Thus figures 5-8 are simply showing the same thing in different ways. However, some of this information is useful, but could be presented
in a much more compact form, e.g. eliminate most of the text, in particular any refererence to particular numberical values, and provide just a brief statement of the various properties
being shown.
Reply: Thank you for the comment, for your kind information, all the parameters given in the figures are related to each other, but from each parameter we can get useful information. For example, from the linear attenuation coefficient we can understand the influence of the density of the glass sample on the attenuation ability of the glass sample against gamma radiation, while from the half value layer (which is a parameter derived from the LAC, we can get information about the thickness needed to decrease the intensity of the radiation by 50% of the original value. Thus, from the LAC and HVL we can get different physical information. However, in this work and as per your valuable suggestion, we removed some unnecessary information from figs5-8 and we deleted Fig.8 according to your next comment. Also, we removed some the numerical values in the HVL and TVL (Figs. 5 and 6), but we keep some numerical values for the mean free path (Fig.7) for one reason. We believe that it is useful to mention some numerical values for one parameter at least in the paper to help the other researchers in finding some values for the comparisons between their glass samples and other glasses reported in the literature. For this reason, we kept some numerical values related to Fig.7.
Second, show figure 6 using the same style as in figure 5.
Reply: Thank you for the comment, we revised Fig.6 using the same style as in Fig.5
Third, is figure 8 necessary since figure 1 already has shown the LAC?
Reply: We agree with you in this comment and we removed Fig.8
Four: Summarize the findings at the end of this section or in the conclusion.
Reply: We summarized the main findings in this work in the conclusion
Figure 9: Very nice. This is probably the message that is most important to the reader.
Reply: Thank you very much for your nice comment
Reviewer 2 Report
This manuscript is poorly written; important sections such as methods, results and discussion are explained abstractly and ambiguously, thus making the manuscript hard to comprehend. Secondly, extensive editing of the English language and style is required for this manuscript. Based on the above-mentioned grounds, this manuscript must be rejected.
Author Response
This manuscript is poorly written; important sections such as methods, results and discussion are explained abstractly and ambiguously, thus making the manuscript hard to comprehend. Secondly, extensive editing of the English language and style is required for this manuscript. Based on the above-mentioned grounds, this manuscript must be rejected.
Reply: Thank you for your comment. Since you mentioned (extensive editing of the English language and style is required for this manuscript.), we revised the language of the paper using English language editing by MDPI. Below is the certificate
Moreover, we extended the method used for the calculation of the different parameters in this work, and we added the essential equations so the readers can understand the method used in this work for the determination of the parameters. Also, we extended the discussion of the results and added more details.

Reviewer 3 Report
Introduction: the manuscript discussing (theoretically) the radiation shielding properties of a glass system using PHY-X software after photon absorption ( some energy range) with a range of and several parameters are selected and examined, such as LAC, MAC, Zeff, HVL, and so on,
Comments:
Figure 2, LAC results for energies 0.6617 and 0.637. 1- All the figures without error bars? The authors recommended adding error bars for most of the figures and discuss the possibility of overlapped data, e. g.
2- the results of the relation of LAC and MCA in Fig. 2 and 3 are stated and not discussed; only the tendency of the results are mentioned? Discussion for the results are needed, not only the observations from the figures?
3- line 137, " .... with BLa10 having ..........Advantage" conclusion without details discussion for the statement.
4- Fig. 3, there is almost no difference for the MCA for BLaxx compounds data for energies 0.511 ~1.275? A comment is needed.
5- the MFP as a function of the density in Fig. 7 for energies 0.6617 and 0.637 are overlapped. A comment is needed, or removing one of the data can remove the missleading presentation.
6- Comments on the attitude of energy 1.275 in Fig. 7 are also needed.
7- Fig. 9 comments on the observation of high HVL data for the commercial RS-253-G18 is needed compared to other compounds.
8- I expected the authors to give a vision of the possibility to fabricate the best composition in the form of a sample and subjected it to a real experimental test to verify the calculations with the experimental results.
9- the English quality of the manuscript is not sufficient for the journal; proofreading is needed.
Author Response
Comments and Suggestions for Authors
Introduction: the manuscript discussing (theoretically) the radiation shielding properties of a glass system using PHY-X software after photon absorption ( some energy range) with a range of and several parameters are selected and examined, such as LAC, MAC, Zeff, HVL, and so on,
Comments:
Figure 2, LAC results for energies 0.6617 and 0.637. 1- All the figures without error bars? The authors recommended adding error bars for most of the figures and discuss the possibility of overlapped data, e. g.
2- the results of the relation of LAC and MCA in Fig. 2 and 3 are stated and not discussed; only the tendency of the results are mentioned? Discussion for the results are needed, not only the observations from the figures?
Reply: Thank you very much for your constructive comment, we added more explanations about the LAC and MAC. For example, we added the reason for the increasing in the LAC and MAC values with adding more amount of La2O3. Also, we explained the high LAC at low energies and added the reason for the small change in the LAC and MAC at higher energies and added suitable reference.
3- line 137, " .... with BLa10 having ..........Advantage" conclusion without details discussion for the statement.
Reply: Thank you very much for your comment, we added a brief discussion about this statement.
4- Fig. 3, there is almost no difference for the MCA for BLaxx compounds data for energies 0.511 ~1.275? A comment is needed.
Reply: Thank you very much for your comment, we added the reason for this observation according to the Compton scattering which has a very weak dependence on the medium.
5- the MFP as a function of the density in Fig. 7 for energies 0.6617 and 0.637 are overlapped. A comment is needed, or removing one of the data can remove the missleading presentation.
Reply: We removed the data for the energy 0.6617 MeV
6- Comments on the attitude of energy 1.275 in Fig. 7 are also needed.
Reply: Thank you for your comment, we added the explanation for the high MFP at 1.275 MeV and mentioned that the penetrating ability of the high energy photons is very high and thus we needed a relatively thick glass to attenuate these high energy photons
7- Fig. 9 comments on the observation of high HVL data for the commercial RS-253-G18 is needed compared to other compounds.
Reply: We added the following sentence as an explanation to the high MFP for RS-253-G18:
‘’The high MFP for the RS-253-G18 is due to the low density of this glass comparing to our selected glasses, where the density of RS-253-G18 is 2.53 g/cm3, while the density of our glasses varied between 3.01 and 3.76 g/cm3 ’’
8- I expected the authors to give a vision of the possibility to fabricate the best composition in the form of a sample and subjected it to a real experimental test to verify the calculations with the experimental results.
9- the English quality of the manuscript is not sufficient for the journal; proofreading is needed.
We revised the language of the paper using English language editing by MDPI. Below is the certificate

Round 2
Reviewer 1 Report
The third version of this manuscript that I have read still suffers from what I shall refer to as "excess padding". By this I mean, information is presented that isn't necessary for the reader to know in
order to understand the major findings, e.g., listing of numerical values to several decimal places when values of sufficient accuracy could be extracted from the figures, and very similar
information is shown in several different ways, e.g., LAC, HVL and TVL. This tends to hide the major findings plus I estimate that it lengthens the manuscript by roughly 25-30%.
The primary examples are contained in the list below. From the author's previous response, they think all this information is valuable and needs to be presented; my
opinion is that it is not. For this reason, I cannot recommend publication in the present form.
line 102-103: symbol in line different from symbol in formula. i^th incorrectly written. mu and rho need to be defined.
line 129: symbol in text different from symbol in formula.
lines 137-145: As stated in my previous review, this manuscript is heavily padded by providing unnecessary information, namely numerical values for the energies and the LAC. These
values are either available from the figures or are not necessary for the reader to know to 3 decimal places. Delete the sentences on lines 137-141 and 142-143 and modify/shorten the sentence
on lines 144-145 appropriately.
lines 181-193: Another example of unnecessary padding of information, e.g., the only important information here is that Zeff increases with La content and decreases with energy with the
sharp decrease attributed to Compton scattering. What value the exact Zeff numbers have is not obvious to me.
Fig. 6 and associated text (lines 214-224): These can be elliminated by stating that from equations 3 and 4 the TVL values are 3.3 times larger than the HVL shown in Fig. 5.
lines 231-234: In accordance with statements above, this sentence can be deleted. Same for sentence on lines 239-241.
line 241: "great spike" implies a fast increase followed by a fast decrease. "large change" may be a better description.
lines 249-260: This provides the essential information about the figure in a compact form that unfortunately is lacking for the previous figures.
line 270-271: Just give the range of Zeff similar to what is done in the previous sentences.
Author Response
The third version of this manuscript that I have read still suffers from what I shall refer to as "excess padding". By this I mean, information is presented that isn't necessary for the reader to know in order to understand the major findings, e.g., listing of numerical values to several decimal places when values of sufficient accuracy could be extracted from the figures, and very similar information is shown in several different ways, e.g., LAC, HVL and TVL. This tends to hide the major findings plus I estimate that it lengthens the manuscript by roughly 25-30%.
The primary examples are contained in the list below. From the author's previous response, they think all this information is valuable and needs to be presented; my opinion is that it is not. For this reason, I cannot recommend publication in the present form.
line 102-103: symbol in line different from symbol in formula. i^th incorrectly written. mu and rho need to be defined.
Reply: Thank you for your comment, we revised it.
line 129: symbol in text different from symbol in formula.
Reply: sorry, but we didn’t find any symbol in line 129, and we checked the symbols in the text and revised some symbols to be consistent with the formula
lines 137-145: As stated in my previous review, this manuscript is heavily padded by providing unnecessary information, namely numerical values for the energies and the LAC. These
values are either available from the figures or are not necessary for the reader to know to 3 decimal places. Delete the sentences on lines 137-141 and 142-143 and modify/shorten the sentence on lines 144-145 appropriately.
Reply: Thank you for this important and constructive comment, we deleted these unnecessary information.
lines 181-193: Another example of unnecessary padding of information, e.g., the only important information here is that Zeff increases with La content and decreases with energy with the
sharp decrease attributed to Compton scattering. What value the exact Zeff numbers have is not obvious to me.
Reply: Thank you for this important and constructive comment, we deleted the unnecessary information. However, we would like to explain why we include these numerical values. We believe that this field is very interesting for many other researchers and no enough data (LAC, HVL, Zeff….etc) for different glasses are available in the literature, so we tried to include some numerical values and these values will help other researchers to find some reported data so they can compare their experimental and theoretical findings with our reported data. However, as per your valuable suggestion, we deleted these numerical values.
Fig. 6 and associated text (lines 214-224): These can be elliminated by stating that from equations 3 and 4 the TVL values are 3.3 times larger than the HVL shown in Fig. 5.
Reply: We agree with you that the HVL and TVL are almost similar parameters, so we deleted equation 4 and Fig.6 and the associated text (lines 214-224)
lines 231-234: In accordance with statements above, this sentence can be deleted. Same for sentence on lines 239-241.
Reply: We agree with you in this comment and deleted these sentences in lines 231-234 and 239-241
line 241: "great spike" implies a fast increase followed by a fast decrease. "large change" may be a better description.
Reply: We changed "great spike" to "large change"
lines 249-260: This provides the essential information about the figure in a compact form that unfortunately is lacking for the previous figures.
Reply: Sorry for the typo, the figure was already in the manuscript, but by mistake we wrote it as fig.9. Now, it is Fig.7 in the revised version
line 270-271: Just give the range of Zeff similar to what is done in the previous sentences.
Reply: Revised as per your suggestion
Reviewer 2 Report
This manuscript can be accepted for publication.
Author Response
Thank you for your efforts in improving this paper
Reviewer 3 Report
Include the error bars for the data given in all the figures included in the manuscript is mandatory for the discussion understanding.
Author Response
Thank you for your comment, We would like to explain that paper is based on theoretical calculations using Phy-X software, and no experimental data are presented, so we it is not suitable to include the error bars in the figures